# Neural Dynamic Policies
# for End-to-End Sensorimotor Learning

**Shikhar Bahl**[*]
CMU

**Mustafa Mukadam**
FAIR

**Abhinav Gupta**
CMU

**Deepak Pathak**
CMU

## Abstract

The current dominant paradigm in sensorimotor control, whether imitation or reinforcement learning, is to train policies directly in raw action spaces such as torque, joint angle, or end-effector position. This forces the agent to make decision at each point in training, and hence, limit the scalability to continuous, high-dimensional, and long-horizon tasks. In contrast, research in classical robotics has, for a long time, exploited dynamical systems as a policy representation to learn robot behaviors via demonstrations. These techniques, however, lack the flexibility and generalizability provided by deep learning or deep reinforcement learning and have remained under-explored in such settings. In this work, we begin to close this gap and embed dynamics structure into deep neural network-based policies by reparameterizing action spaces with differential equations. We propose Neural Dynamic Policies (NDPs) that make predictions in trajectory distribution space as opposed to prior policy learning methods where action represents the raw control space. The embedded structure allow us to perform end-to-end policy learning under both reinforcement and imitation learning setups. We show that NDPs achieve better or comparable performance to state-of-the-art approaches on many robotic control tasks using both reward-based training and demonstrations. Project video and code are available at: https://shikharbahl.github.io/neural-dynamic-policies/.

## 1 Introduction

Consider an embodied agent tasked with throwing a ball into a bin. Not only does the agent need to decide where and when to release the ball, but also reason about the whole trajectory that it should take such that the ball is imparted with the correct momentum to reach the bin. This form of reasoning is necessary to perform many such everyday tasks. Common methods in deep learning for robotics tackle this problem either via imitation or reinforcement. However, in most cases, the agent's policy is trained in raw action spaces like torque, joint angle, or end-effector position, which forces the agent to make decisions at each time step of the trajectory instead of making decisions in the trajectory space itself (see Figure 1). But then how do we reason about trajectories as actions?

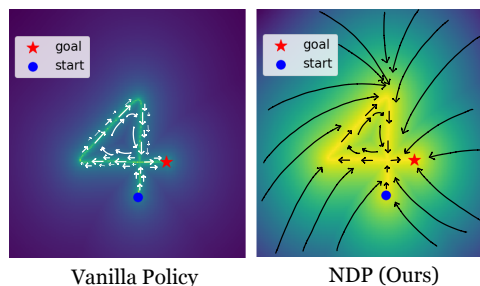

Vanilla Policy      NDP (Ours)

Figure 1: Vector field induced by NDPs. The goal is to draw the planar digit 4 from the start position. The dynamical structure in NDP induces a smooth vector field in trajectory space. In contrast, a vanilla policy has to reason individually in different parts.

---

[*]Correspondence to: sbahl2@cs.cmu.edu

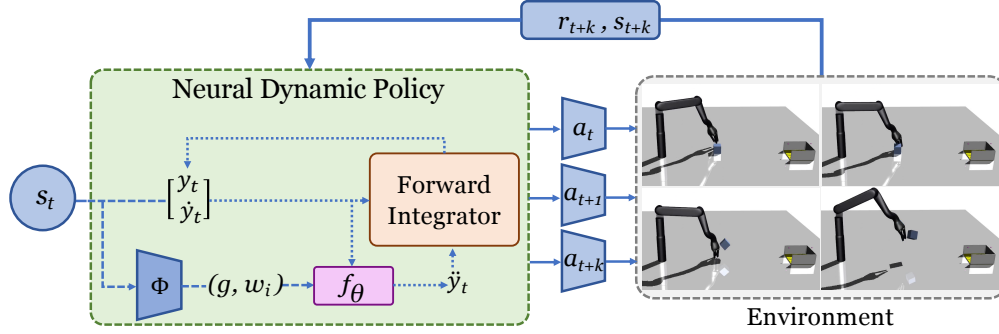

Figure 2: Given an observation from the environment, $s_t$, our Neural Dynamic Policy generates parameters $w$ (weights of basis functions) and $g$ (goal for the robot) for a forcing function $f_\theta$. An open loop controller then uses this function to output a set of actions for the robot to execute in the environment, collecting future states and rewards to train the policy.

A good trajectory parameterization is one that is able to capture a large set of agent's behaviors or motions while being physically plausible. In fact, a similar question is also faced by physicists while modeling physical phenomena in nature. Several systems in science, ranging from motion of planets to pendulums, are described by differential equations of the form $\ddot{y} = m^{-1}f(y, \dot{y})$, where $y$ is the generalized coordinate, $\dot{y}$ and $\ddot{y}$ are time derivatives, $m$ is mass, and $f$ is force. Can a similar parameterization be used to describe the behavior of a robotic agent? Indeed, classical robotics has leveraged this connection to represent task specific robot behaviors for many years. In particular, dynamic movement primitives (DMP) [20–22, 37] have been one of the more prominent approaches in this area. Despite their successes, DMPs have not been explored much beyond behavior cloning paradigms. This is partly because these methods tend to be sensitive to parameter tuning and aren't as flexible or generalizable as current end-to-end deep network based approaches.

In this work, we propose to bridge this gap by embedding structure of dynamical systems[2] into deep neural network-based policies such that the agent can directly learn in the space of physically plausible trajectory distributions (see Figure 1(b)). **Our key insight is to reparameterize the action space in a deep policy network with nonlinear differential equations corresponding to a dynamical system and train it end-to-end over time in either reinforcement learning or imitation learning setups.** However, this is quite challenging to accomplish, since naively predicting a full arbitrary dynamical system directly from the input, trades one hard problem for another. Instead, we want to prescribe some structure such that the dynamical system itself manifests as a layer in the deep policy that is both, amenable to take arbitrary previous layers as inputs, and is also fully differentiable to allow for gradients to backpropagate.

We address these challenges through our approach, Neural Dynamic Policies (NDPs). Specifically, NDPs allow embedding desired dynamical structure as a layer in deep networks. The parameters of the dynamical system are then predicted as outputs of the preceding layers in the architecture conditioned on the input. **The 'deep' part of the policy then only needs to reason in the lower-dimensional space of building a dynamical system that then lets the overall policy easily reason in the space of trajectories.** In this paper, we employ the aforementioned DMPs as the structure for the dynamical system and show its differentiability, although they only serve as a design choice and can possibly be swapped for a different differentiable dynamical structure, such as RMPs [35].

We evaluate NDPs in imitation as well as reinforcement learning setups. NDPs can utilize high-dimensional inputs via demonstrations and learn from weak supervisory signals as well as rewards. In both setups, NDPs exhibit better or comparable performance to state-of-the-art approaches.

## 2 Modeling Trajectories with Dynamical Systems

Consider a robotic arm exhibiting a certain behavior to accomplish some task. Given a choice of coordinate system, such as either joint-angles or end-effector position, let the state of the robot be $y$, velocity $\dot{y}$ and acceleration $\ddot{y}$. In mechanics, Euler-Lagrange equations are used to derive

the equations of motion as a general second order dynamical system that perfectly captures this behavior [39, Chapter 6]. It is common in classical robotics to represent movement behaviors with such a dynamical system. Specifically, we follow the second order differential equation structure imposed by Dynamic Movement Primitives [22, 28, 37]. Given a desired goal state $g$, the behavior is represented as:

$$\ddot{y} = \alpha(\beta(g - y) - \dot{y}) + f(x), \tag{1}$$

where $\alpha, \beta$ are global parameters that allow critical damping of the system and smooth convergence to the goal state. $f$ is a non-linear forcing function which captures the shape of trajectory and operates over $x$ which serves to replace time dependency across trajectories, giving us the ability to model time invariant tasks, e.g., rhythmic motions. $x$ evolves through the first-order linear system:

$$\dot{x} = -a_x x \tag{2}$$

The specifics of $f$ are usually design choices. We use a sum of weighted Gaussian radial basis functions [22] shown below:

$$f(x, g) = \frac{\sum \psi_i w_i}{\sum \psi_i} x(g - y_0), \quad \psi_i = e^{(-h_i(x - c_i)^2)} \tag{3}$$

where $i$ indexes over $n$ which is the number of basis functions. Coefficients $c_i = e^{\frac{-i\alpha_x}{n}}$ are the horizontal shifts of each basis function, and $h_i = \frac{n}{c_i}$ are the width of each of each basis function. The weights on each of the basis functions $w_i$ parameterize the forcing function $f$. This set of nonlinear differential equations induces a smooth trajectory distribution that acts as an attractor towards a desired goal, see Figure 1(right). We now discuss how to combine this dynamical structure with deep neural network based policies in an end-to-end differentiable manner.

## 3 Neural Dynamic Policies (NDPs)

We condense actions into a space of trajectories, parameterized by a dynamical system, while keeping all the advantages of a deep learning based setup. We present a type of policy network, called Neural Dynamic Policies (NDPs) that given an input, image or state, can produce parameters for an embedded dynamical structure, which reasons in trajectory space but output raw actions to be executed. Let the unstructured input to robot be $s$, (an image or any other sensory input), and the action executed by the robot be $a$. We describe how we can incorporate a dynamical system as a differentiable layer in the policy network, and how NDPs can be utilized to learn complex agent behaviors in both imitation and reinforcement learning settings.

### 3.1 Neural Network Layer Parameterized by a Dynamical System

Throughout this paper, we employ the dynamical system described by the second order DMP equation (1). There are two key parameters that define what behavior will be described by the dynamical system presented in Section 2: basis function weights $w = \{w_1, \ldots, w_i, \ldots, w_n\}$ and goal $g$. NDPs employ a neural network $\Phi$ which takes an unstructured input $s$[3] and predicts the parameters $w, g$ of the dynamical system. These predicted $w, g$ are then used to solve the second order differential equation (1) to obtain system states $\{y, \dot{y}, \ddot{y}\}$. Depending on the difference between the choice of robot's coordinate system for $y$ and desired action $a$, we may need an inverse controller $\Omega(.)$ to convert $y$ to $a$, i.e., $a = \Omega(y, \dot{y}, \ddot{y})$. For instance, if $y$ is in joint angle space and $a$ is a torque, then $\Omega(.)$ is the robot's inverse dynamics controller, and if $y$ and $a$ both are in joint angle space then $\Omega(.)$ is the identity function.

As summarized in Figure 2, neural dynamic policies are defined as $\pi(a|s;\theta) \triangleq \Omega(\texttt{DE}(\Phi(s;\theta)))$ where $\texttt{DE}(w, g) \rightarrow \{y, \dot{y}, \ddot{y}\}$ denotes solution of the differential equation (1). The forward pass of $\pi(a|s)$ involves solving the dynamical system and backpropagation requires it to be differentiable. We now show how we differentiate through the dynamical system to train the parameters $\theta$ of NDPs.

## 3.2 Training NDPs by Differentiating through the Dynamical System

To train NDPs, estimated policy gradients must flow from $a$, through the parameters of the dynamical system $w$ and $g$, to the network $\Phi(s; \theta)$. At any time $t$, given the previous state of robot $y_{t-1}$ and velocity $\dot{y}_{t-1}$ the output of the DMP in Equation (1) is given by the acceleration

$$\ddot{y}_t = \alpha(\beta(g - y_{t-1}) - \dot{y}_{t-1} + f(x_t, g) \tag{4}$$

Through Euler integration, we can find the next velocity and position after a small time interval $dt$

$$\dot{y}_t = \dot{y}_{t-1} + \ddot{y}_{t-1}dt, \quad y_t = y_{t-1} + \dot{y}_{t-1}dt \tag{5}$$

In practice, this integration is implemented in $m$ discrete steps. To perform a forward pass, we unroll the integrator for $m$ iterations starting from initial $\dot{y}_0, \ddot{y}_0$. We can either apply all the $m$ intermediate robot states $y$ as action on the robot using inverse controller $\Omega(.)$, or equally sub-sample them into $k \in \{1, m\}$ actions in between, where $k$ is NDP rollout length. This frequency of sampling could allow robot operation at a much higher frequency (.5-5KHz) than the environment (usually 100Hz). The sampling frequency need not be same at training and inference as discussed further in Section 3.5.

Now we can compute gradients of the trajectory from the DMP with respect to $w$ and $g$ using Equations (3)-(5) as follows:

$$\frac{\partial f(x_t, g)}{\partial w_i} = \frac{\psi_i}{\sum_j \psi_j}(g - y_0)x_t, \quad \frac{\partial f(x_t, g)}{\partial g} = \frac{\psi_j w_j}{\sum_j \psi_j}x_t \tag{6}$$

Using this, a recursive relationship follows between, (similarly to the one derived by Pahic et al. [26]) $\frac{\partial y_t}{\partial w_i}, \frac{\partial y_t}{\partial g}$ and the preceding derivatives of $w_i$, $g$ with respect to $y_{t-1}, y_{t-2}, \dot{y}_{t-1}$ and $\dot{y}_{t-2}$. Complete derivation of equation (6) is given in appendix.

We now discuss how NDPs can be leveraged to train policies for imitation learning and reinforcement learning setups.

## 3.3 Training NDPs for Imitation (Supervised) Learning

Training NDPs in imitation learning setup is rather straightforward. Given a sequence of input $\{s, s', \dots\}$, NDP's $\pi(s; \theta)$ outputs a sequence of actions $a, a' \dots$. In our experiments, $s$ is the high dimensional image input. Let the demonstrated action sequence be $\tau_{\text{target}}$, we just take a loss between the predicted sequence as follows:

$$\mathcal{L}_{\text{imitation}} = \sum_s ||\pi(s) - \tau_{\text{target}}(s)||^2 \tag{7}$$

The gradients of this loss are backpropagated as described in Section 3.2 to train the parameters $\theta$.

## 3.4 Training NDPs for Reinforcement Learning

We now show how an NDP can be used as a policy, $\pi$ in the RL setting. As discussed in Section 3.2, NDP samples k actions for the agent to execute in the environment given input observation $s$. One could use any underlying RL algorithm to optimize the expected future returns. In this paper, we use Proximal Policy Optimization (PPO) [38] and treat $a$ independently when computing the policy gradient for each step of the NDP rollout and backprop via a reinforce objective.

There are two choices for value function critic $V^\pi(s)$: either predict a single common value function for all the actions in the $k$-step rollout or predict different critic values for each step in the NDP rollout sequence. We found that the

---

**Algorithm 1** Training NDPs for RL

**Require:** Policy $\pi$, $k$ NDP rollout length, $\Omega$ low-level inverse controller
  **for** $1, 2, \dots$ episodes **do**
    **for** $t = 0, k, \dots$, until end of episode **do**
      $w, g = \Phi(s_t)$
      Robot $y_t, \dot{y}_t$ from $s_t$ (pos, vel)
      **for** $m = 1, \dots, M$ (integration steps) **do**
        Estimate $\dot{x}_m$ via (2) and update $x_m$
        Estimate $\ddot{y}_{t+m}, \dot{y}_{t+m}, y_{t+m}$ via (4), (5)
        $a = \Omega(y_{t+m}, y_{t+m-1})$
        Apply action $a$ to get $s'$
        Store transition $(s, a, s', r)$
      **end for**
      Compute Policy gradient $\nabla_\theta$
      $\theta \leftarrow \theta + \eta \nabla_\theta J$
    **end for**
  **end for**

---

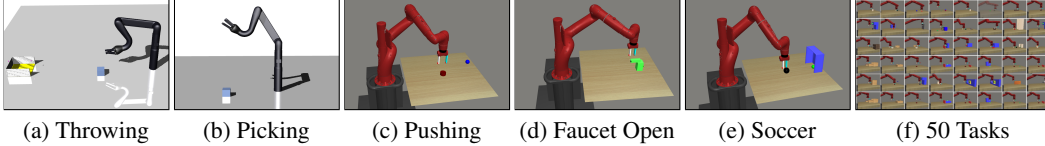

| (a) Throwing | (b) Picking | (c) Pushing | (d) Faucet Open | (e) Soccer | (f) 50 Tasks |

Figure 3: Environment snapshot for different tasks considered in experiments. (a,b) Throwing and picking tasks are adapted from [17] on the Kinova Jaco arm. (c-f) Remaining tasks are adapted from [46]

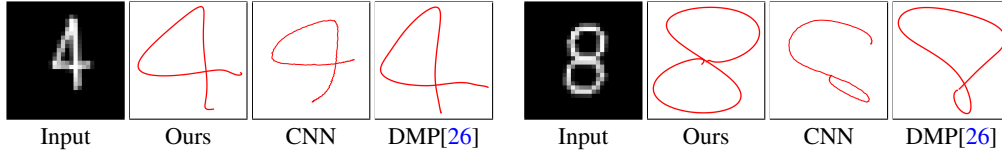

| Input | Ours | CNN | DMP[26] | Input | Ours | CNN | DMP[26] |

Figure 4: Imitation (supervised) learning results on held-out test images of digit writing task. Given an input image (left), the output action is the end-effector position of a planar robot. All methods have the same neural network architecture for fair comparison. We find that the trajectories predicted by NDPs (ours) are dynamically smooth as well as more accurate than both baselines.

latter works better in practice. We call this *multi-action critic architecture* and predict $k$ different estimates of value using $k$-heads on top of critic network. Later, in the experiments we perform ablations over the choice of $k$. To further create a strong baseline comparison, as we discuss in Section 4, we also design and compare against a variant of PPO that predicts multiple actions using our multi-action critic architecture.

Algorithm 1 provides a summary of our method for training NDPs with policy gradients. We only show results of using NDPs with on-policy RL (PPO), however, NDPs can also be adapted similarly to off-policy methods.

### 3.5 Inference in NDPs

In the case of inference, our method uses the NDP policy $\pi$ once every $k$ environment steps, hence requires $k$-times fewer forward passes as actions applied to the robot. While reducing the inference time in simulated tasks may not show much difference, in real world settings, where large perception systems are usually involved, reducing inference time can help decrease overall time costs. Additionally, deployed real world systems may not have the same computational power as many systems used to train state-of-the-art RL methods on simulators, so inference costs end up accumulating, thus a method that does inference efficiently can be beneficial. Furthermore, as discussed in Section 3.2, the rollout length of NDP can be more densely sampled at test-time than at training allowing the robot to produce smooth and dynamically stable motions. Compared to about 100Hz frequency of the simulation, our method can make decisions an order of magnitude faster (at about 0.5-5KHz) at inference.

## 4 Experimental Setup

**Environments** To test our method on dynamic environments, we took existing torque control based environments for Picking and Throwing [17] and modified them to enable joint angle control. The robot is a 6-DoF Kinova Jaco Arm. In Throwing, the robot tosses a cube into a bin, and in Picking, the robot picks up a cube and lifts it as high as possible. To test on quasi-static tasks, we use Pushing, Soccer, Faucet-Opening from the Meta-World [46] task suite, as well as a setup that requires learning all 50 tasks (MT50) jointly (see Figure 3). These Meta-World environments are all in end-effector position control settings and based on a Sawyer Robot simulation in Mujoco [43]. In order to make the tasks more realistic, all environments have some degree of randomization. Picking and Throwing have random starting positions, while the rest have randomized goals.

**Baselines** We use PPO [38] (PPO) as the underlying optimization algorithm for NDPs and all the other baselines compared in the reinforcement learning setup. The first baseline is the PPO algorithm itself without the embedded dynamical structure. Further, as mentioned in the Section 3.2, NDP

is able to operate the robot at a much higher frequency than the world. Precisely, it's frequency is $k$-times higher where $k$ is the NDP rollout length (described in Section 3.2). Even though the robot moves at a higher frequency, the environment/world state is only observed at normal rate, i.e., once every $k$ robot steps and the reward computation at the intermediate $k$ steps only use stale environment/world state from the first one of the $k$-steps. Hence, to create a stronger baseline that can also operate at higher frequency, we create a "PPO-multi" baseline that predicts multiple actions and also uses our *multi-action critic* architecture as described in Section 3.4. All methods are compared in terms of performance measured against the environment sample states observed by the agent. In addition, we also compare to Variable Impedance Control in End-Effector Space (VICES) [24] and Dynamics-Aware Embeddings (Dyn-E) [45] . VICES learns to output parameters of a PD controller or an Impedance controller directly. Dyn-E, on the other hand, using forward prediction based on environment dynamics, learns a lower dimensional action embedding.

## 5 Evaluation Results: NDPs for Imitation and Reinforcement Learning

We validate our approach on Imitation Learning and RL tasks in order to ascertain how our NDP compares to state-of-the-art methods. We investigate: a) Does dynamical structure in NDPs help in learning from demonstrations in imitation learning setups?; b) How well do NDPs perform on dynamic and quasi-static tasks in deep reinforcement learning setups compared to the baselines?; c) How sensitive is the performance of NDPs to different hyper-parameter settings?

### 5.1 Imitation (Supervised) Learning

To evaluate NDPs in imitation learning settings we train an agent to perform various control tasks. We evaluate NDPs on the Mujoco [43] environments discussed in Section 4 (Throwing, Picking, Pushing, Soccer and Faucet-Opening). Experts are trained using PPO [38] and are subsequently used to collect trajectories. We train an NDP via the behaviour cloning procedure described in Section 3.3, on the collected expert data. We compare against a neural network policy (using roughly the same model capacity for both). Success rates in Table 1 indicate that NDPs show superior performance on a wide variety of control tasks.

| Method | NN | NDP (ours) |
|--------|------|-----------|
| Throw  | $0.528 \pm 0.262$ | $\mathbf{0.642 \pm 0.246}$ |
| Pick   | $\mathbf{0.672 \pm 0.074}$ | $0.408 \pm 0.058$ |
| Push   | $0.002 \pm 0.004$ | $\mathbf{0.208 \pm 0.049}$ |
| Soccer | $0.885 \pm 0.016$ | $\mathbf{0.890 \pm 0.010}$ |
| Faucet | $0.532 \pm 0.231$ | $\mathbf{0.790 \pm 0.059}$ |

Table 1: Imitation (supervised) learning results (success rates between 0 and 1) on Mujoco [43] environments. We see that NDP outperforms the neural network baseline in many tasks.

In order to evaluate the ability of NDPs to handle complex visual data, we perform the task of learning to write digits using a 2D end-effector. The goal is to train a planar robot to trace the digit given its image as input. The output action is the robot's end-effector position, and supervision is obtained by treating ground truth trajectories as demonstrations. We compare NDPs to a regular behavior cloning policy parametrized by a CNN and the prior approach which maps image to DMP parameters [26] (dubbed, CNN-DMP). CNN-DMP [26] trains a single DMP for the whole trajectory and requires supervised demonstrations, which is in contrast to NDPs can generate multiple DMPs across time and can be used in RL setup as well. However, for a fair comparison, we compare both methods apples-to-apples with single DMP for whole trajectory, i.e., $k = 300$.

Qualitative examples are in Figure 4 and quantitative results in Table 2 report the mean loss between output trajectory and ground truth. NDP outperforms both CNN and CNN-DMP [26] drastically. Our method also produces much higher quality and smoother reconstructions as shown in Figure 4. Results show that our method can efficiently capture dynamic motions in a supervised setting, while learning from visual data.

| Method | Train | Test (held-out) |
|--------|-------|-----------------|
| CNN          | $10.42 \pm 5.26$ | $10.59 \pm 4.63$ |
| CNN-DMP [26] | $9.44 \pm 4.59$  | $8.46 \pm 8.45$  |
| NDP (ours)   | $\mathbf{0.70 \pm 0.36}$ | $\mathbf{0.74 \pm 0.34}$ |

Table 2: Imitation learning on digit writing task. We report the mean loss across 10 digit classes. The input is the image of the digit to be written and action output is the end-effector position of robot. Our method significantly outperforms the baseline.

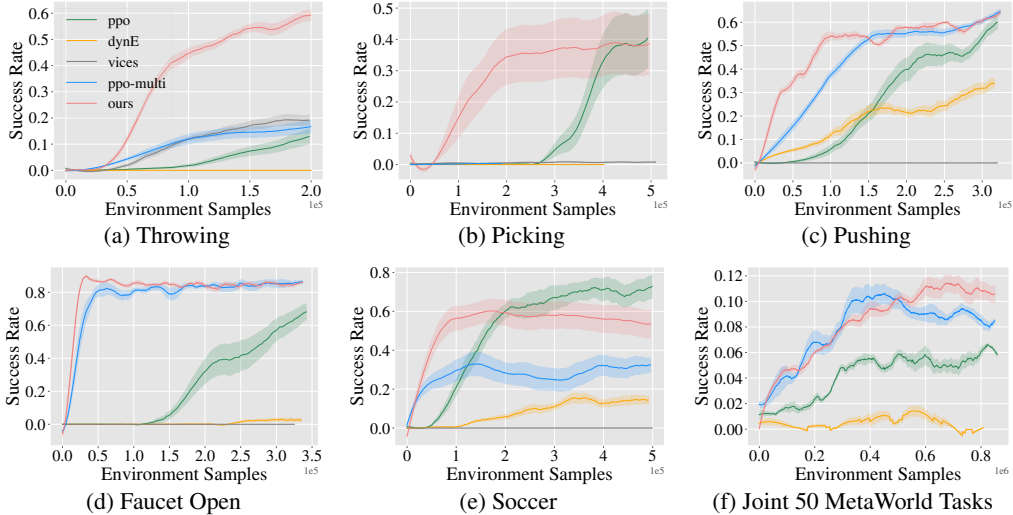

Figure 5: Evaluation of reinforcement learning setup for continuous control tasks. Y axis is success rate and X axis is number of environment samples. We compare to PPO [38], a multi-action version of PPO, VICES [24] and DYN-E [45]. The dynamic rollout for NDP & PPO-multi is $k = 5$.

## 5.2 Reinforcement Learning

In contrast to imitation learning where the rollout length of NDP is high ($k = 300$), we set $k = 5$ in RL because the reward becomes too sparse if $k$ is very large. We compare the success rate of our method with that of the baseline methods PPO, a version of PPO which outputs multiple actions ($k = 5$), VICES and DYN-E.

As shown in In Figure 5, our method NDP sees gains in both efficiency and performance in most tasks. In Soccer, PPO reaches a higher final performance, but our method shows twice the efficiency at small loss in performance. The final task of training jointly across 50 Meta-World tasks is too hard for all methods. Nevertheless, our NDP attains slightly higher absolute performance than baseline but doesn't show efficiency gains over baselines.

PPO-multi, a multi-action algorithm based on our proposed multi-action critic setup tends to perform well in some case (Faucet Opening, Pushing etc) but is inconsistent in its performance across all tasks and fails completely at times, (Picking etc.). Our method also outperforms prior state-of-the-art methods that re-paremeterize action spaces, namely, VICES [24] and Dyn-E [45]. VICES is only slightly successful in tasks like throwing, since a PD controller can efficiently solve the task, but suffer in more complex settings due to a large action space dimensionality (as it predicts multiple quantities per degree of freedom). Dyn-E, on the other hand, performs well on tasks such as Pushing, or Soccer, which have simpler dynamics and contacts, but fails to scale to more complex environments.

Through these experiments, we show the diversity and versatility of NDP, as it has a strong performance across different types of control tasks. NDP outperforms baselines in both dynamic (throwing) and static tasks (pushing) while being able to learn in a more data efficient manner. It is able to reason in a space of physically meaningful trajectories, but it does not lose the advantages and flexibility of other policy setups have.

### 5.2.1 Ablations for NDPs in Reinforcement Learning Setup

We aim to understand how design choices affect the RL performance of NDP. We run comparisons on the pushing task, varying the number of basis functions $N$ (in the set $\{2, 6, 10, 15, 20\}$), DMP rollout lengths (in set $\{3, 5, 7, 10, 15\}$), number of integration steps (in set $\{15, 25, 35, 45\}$), as well as different basis functions: Gaussian RBF (standard), $\psi$ defined in Equation (3), a liner map $\psi(x) = x$, a multiquadric map: $\psi(x) = \sqrt{1 + (\epsilon x)^2}$, a inverse quadric map $\psi(x) = \frac{1}{1+(\epsilon x)^2}$, and an inverse multiquadric map: $\psi(x) = \frac{1}{\sqrt{1+(\epsilon x)^2}}$.

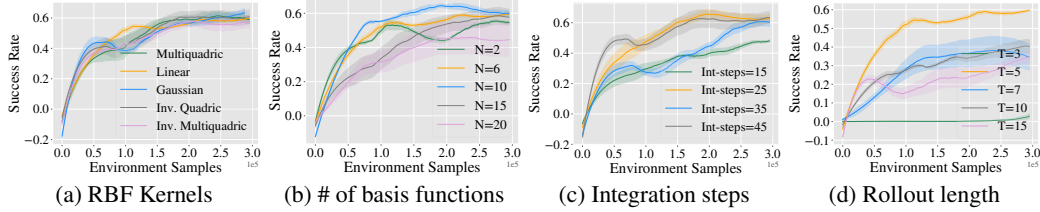

| (a) RBF Kernels | (b) # of basis functions | (c) Integration steps | (d) Rollout length |

Figure 6: Ablation of NDPs with respect to different hyperparameters in the RL setup (pushing). We ablate different choices of radial basis functions in (a). We ablate across number of basis functions, integration steps, and length of the NDP rollout in (b,c,d). Plots indicate that NDPs are fairly stable across a wide range of choices.

Additionally, we investigate the effect of different NDP components on its performance. To this end, we ablate a setting where only $g$ (the goal) is learnt while the radial basis function weights (the forcing function) are 0 (we call this setting 'only-g'). We also ablate a version of NDP that learns the global constant $\alpha$ (from Equation 4), in addition to the other parameters ($g$ and $w$).

Figure 6 shows results from ablating different NDP parameters. Varying $N$ (number of basis functions) controls the shape of the trajectory taken by the agent. A small $N$ may not have the power to represent the nuances of the motion required, while, a big $N$ may make the parameter space too large to learn efficiently. We see that number of integration steps do not have a large effect on performance, similarly to the type of radial basis function. Most radial basis functions generally have similar interpolation and representation abilities. We see that $k = 3$ (the length of each individual rollout within NDP) has a much lower performance due to the fact that 3 steps cannot capture the smoothness or intricacies of a trajectory. Overall, we mostly find that NDP is robust to design choices. Figure 7 shows that the current formulation of NDP outperforms the one where $\alpha$ is learnt. We also observe that setting the forcing term to 0 (only learning the goal, $g$) is significantly less sample efficient than NDPs while converging to a slightly lower asymptotic performance.

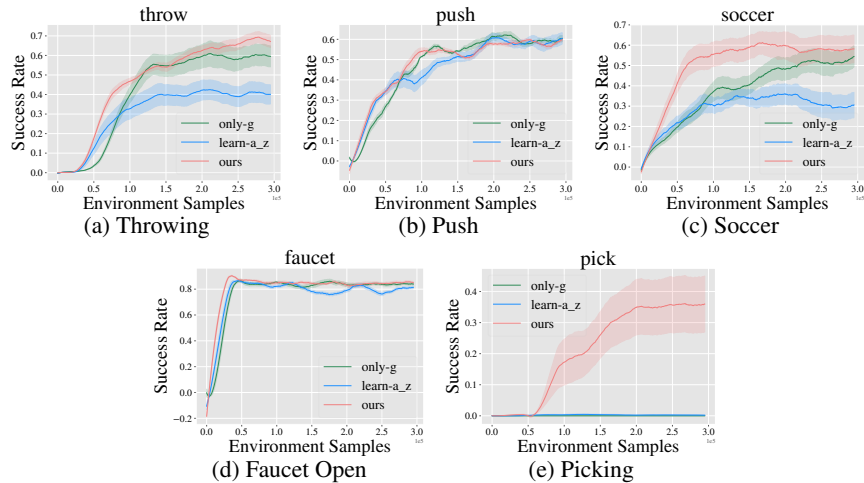

| (a) Throwing | (b) Push | (c) Soccer |

| (d) Faucet Open | (e) Picking |

Figure 7: Ablations for different NDP design choices. The first entails NDP also learning the parameter $\alpha$ (shown as $a_z$). In the second one, $g$ is learnt but not $w_i$, i.e. the forcing function is 0 ('only-g'). Results indicate that NDP outperforms both these settings.

## 6 Related Work

**Dynamic Movement Primitives** Previous works have proposed and used Dynamic Movement Primitives (DMP) [22, 32, 37] for robot control. Work has been done in representing dynamical systems, both as extensions of DMPs [6, 7, 11, 44], and beyond DMPs by learning kernels [19] and designing Riemannian metrics [35]. Learning methods have been used to incorporate environment sensory information into the forcing function of DMPs [33, 41]. DMPs have also been prime candidates to represent primitives when learning hierarchical policies, given the range of motions DMPs can be used for in robotics [13, 23, 29, 40]. Parametrization of DMPs using gaussian processes

has also been proposed to facilitate generalization [31, 44]. Recently, deep learning have also been used to map images to DMPs [26] and to learn DMPs in latent feature space [8]. However most of these works require pre-trained DMPs via expert demonstrations or are only evaluated in the supervised setting. Furthermore, either a single DMP is used to represent the whole task trajectory or the demonstration is manually segmented to learn a different DMP for each segment. In contrast, our proposed NDPs outputs a new dynamical system for each timestep to fit diverse trajectory behaviours across time. Since we embed dynamical structure into the deep network, NDP can flexibly be incorporated not just in visual imitation but also deep reinforcement learning setup, in an end-to-end manner.

**Reparameterized Policy and Action Spaces**   A broader area of work that makes use of action reparameterization is the study of Hierarchical Reinforcement Learning (HRL). Works in the options framework [4, 42] attempt to learn an overarching policy that controls usage of lower-level policies or primitives. Lower-level policies are usually pre-trained therefore require supervision and knowledge of the task beforehand, limiting the generalizability of such methods. For example, Daniel et al. [13], Parisi et al. [27] incorporate DMPs into option-based RL policies, and using a pre-trained DMPs as options. This setup requires re-learning DMPs for different tasks and does not allow the policy the ability to generalize, the policy needs to have access to an extremely large number of DMPs. Action space can also be reparameterized in terms of pre-determined PD controller [47] or learned impedance controller parameters [24]. While this helps for policies to adapt to contact rich behaviors, it does not change the trajectories taken by the robot. In addition, Whitney et al. [45] learn an action embedding based on passive data, however, it does not take environment dynamics or explicit control structure into account.

**Structure in Policy Learning**   Various methods in the field of control and robotics have employed physical knowledge, dynamical systems, optimization, and more general task/environment dynamics to create more structured learning. Works such as [12, 18] are networks constrained through physical properties such as Hamiltonian co-ordinates or Lagrangian Dynamics. However, the scope of these works is limited to toy examples such as a point mass, and are often used for supervised learning. Similarly, other works [25, 30, 34, 36] all employ dynamical systems to model demonstrations, and do not tackle generalization or learning beyond imitation. Fully differentiable optimization problems have also been incorporated as layers inside a deep learning setup [1, 2, 9]. They share the underlying idea of embedding structure in deep networks such that some parameters in the structure can be learned end-to-end, although they haven't been explored in tackling complex robotic control tasks. Further, it is common in RL setups to incorporate planning based on a system model [3, 10, 14–16]. However, this is usually learned from experience and attempts to predict the effects of actions on the environment, and often tend to fail for complex dynamic tasks.

## 7   Discussion

Our method attempts to bridge the gap between classical robotics, control and recent approaches in deep learning and deep RL. We propose a novel re-parameterization of action spaces via Neural Dynamic Policies, a set of policies which impose the structure of a dynamical system on action spaces. We show how this set of policies can be useful for continuous control with RL, and in supervised learning settings. Our method obtains superior results due to its natural imposition of structure and yet it is still generalizable to almost any continuous control environment.

The use of DMPs in this work was a particular design choice within our architecture which allows for any form of dynamical structure that is differentiable. As alluded to in the introduction, other similar representations can be employed in their place. In fact, DMPs are a special case of a general second order dynamical system [5, 35] where the inertia term is identity, and potential and damping functions are defined in a particular manner via first order differential equations with a separate forcing function which captures the complexities of the desired behavior. Given this, one can setup a dynamical structure such that it explicitly models and learns the metric, potential, and damping explicitly. While this brings advantages in better representation it also brings challenges in learning. We leave these directions for future work to explore.

## Acknowledgments

We thank Giovanni Sutanto, Stas Tiomkin and Adithya Murali for fruitful discussions. We also thank Franziska Meier, Akshara Rai, David Held, Mengtian Li, George Cazenavette, and Wen-Hsuan Chu for comments on early drafts of this paper. This work was supported in part by DARPA Machine Common Sense grant and Google Faculty Award to DP.

## Broader Impact

We attempt to create algorithms that empower robotic agents to do complex and long-horizon tasks. However, before our algorithms are deployed in the real world, we must consider how safely our work can interact with humans and their surroundings. We believe that the structured imposed by our method ensures not ensures not only smoother and thus lower risk exploration, but also a larger degree of interpretability of the algorithm. A higher degree of interpretability it is easier reason about how our algorithms will interact with humans, and thus we can operate robots in the wild, in a safer manner.

There are many possible applications of robotics including assembly and manufacturing, medicine, search and rescue, autonomous vehicles and transportation, and slowly moving towards personal robotics. A method that provides safe and efficient real world robotic can have a positive impact by advancing by increasing the quality of assembly lines, minimizing failure in factories, creating more robust search and rescue robots, increasing the flexibility of personal robotics, especially in the case of assistive robots. More efficient automation also has economic benefits and has the potential to save energy resources (using less power to do the same tasks). On the other hand, we must consider negative consequences increased automation, from the the misuse of such technology to the vulnerability of it to external software attacks, to an increase in unemployment.

## Footnotes

[2]Dynamical systems here should not be confused with dynamics model of the agent. We incorporate dynamical differential equations to represent robot's behavioral trajectory and not physical transition dynamics.

[3] robot's state $y$ is not to be confused with environment observation $s$ which contains world as well as robot state (and often velocity). $s$ could be given by either an image or true state of the environment.

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
