[Supplementary Material]

# Supplementary Material:
# Neural Dynamic Policies
# for End-to-End Sensorimotor Learning

**Shikhar Bahl**
CMU

**Mustafa Mukadam**
FAIR

**Abhinav Gupta**
CMU

**Deepak Pathak**
CMU

## 1  Appendix

### 1.1  Qualitative Video Results

Please look at the qualitative difference between NDPs and PPO-multi in generated robot motions in `videos/` folder of supplementary zip file. We found that NDP results look slightly dynamically more stable and smooth in comparison to the baselines. For instance, PPO-multi generates shaky trajectories as can be seen in `ppo_multi_pick.mp4` and `ppo_multi_push.mp4`, while the corresponding NDP (ours) videos in `ndp_pick.mp4` and `ndp_push.mp4` are smoother. This is perhaps due to the embedded dynamical structure in NDPs as all other aspects in PPO-multi and NDP (ours) are compared apples-to-apples. These videos can also be found at: https://shikharbahl.github.io/neural-dynamic-policies/.

### 1.2  Implementation Details

**Hyper-parameters and Design Choices**

We use default parameters and stick as closely as possible to the default code. In multi-action cases (for PPO-multi and NDP), we kept rollout length $k$ fixed for training and at inference time, one can sample arbitrarily high value for $k$ as demanded by the task setup. For reinforcement learning, we kept $k = 5$ because the reward becomes too sparse if $k$ is very large. For imitation learning, this is not an issue, and hence, $k = 300$ for learning from demonstration.

In reinforcement learning setup for NDP, we tried number of basis functions in [4, 5, 6, 7, 10] for each RL task. We fixed the number of integration steps per NDP rollout to 35. We also tried $\alpha$ (as described in Section 2) values in [10, 15, 25]. NDP (ours), PPO [4], PPO-Multi, VICES [3] all use a similar 3-layer fully-connected network of hidden layer sizes of [100, 100] with tanh non-linearities. All these use PPO [4] as the underlying RL optimizer. For Dyn-E [5], we used off-the-shelf architecture because it is based on off-policy RL and doesn't use PPO.

Hyper-parameters for underlying PPO optimization use off-the-shelf without any further tuning from PPO [4] implementation in Kostrikov [2] as follows:

**Environments**

For Picking and Throwing, we adapted tasks from Ghosh et al. [1] (https://github.com/dibyaghosh/dnc. We modified these tasks to enable joint-angle position control. For other RL tasks, we used the Meta-World environment package [6] (https://github.com/rlworkgroup/metaworld). Since VICES [3] operates using torque control, we modified Meta-World environments to support torque-based Impedance control.

| Hyperparameter | Value |
| --- | --- |
| Learning Rate | $3 \times 10^{-4}$ |
| Discount Factor | 0.99 |
| Use GAE | True |
| GAE Discount Factor | 0.95 |
| Entropy Coefficient | 0 |
| Normalized Observations | True |
| Normalized Returns | True |
| Value Loss Coefficient | 0.5 |
| Maximum Gradient Norm | 0.5 |
| PPO Mini-Batches | 32 |
| PPO Epochs | 10 |
| Clip Parameter | 0.1 |
| Optimizer | Adam |
| Batch Size | 2048 |
| RMSprop optimizer epsilon | $10^{-5}$ |

Further, as mentioned in the Section 3.4 and 4, NDP and PPO-multi are able to operate the robot at a higher frequency than the world. Precisely, frequency is $k$-times higher where $k = 5$ is the NDP rollout length (described in Section 3.2). Even though the robot moves at higher frequency, the environment/world state is only observed at normal rate, i.e., once every $k$ robot steps and the reward computation at the intermediate $k$ steps only use stale environment/world state from the first one of the $k$-steps. For instance, if the robot is pushing a puck, the reward is function of robot as well as puck's location. The robot will knows its own position at every policy step but will have access to stale value of puck's location only from actual environment step (sampled at a lower frequency than policy steps, specifically 5x less). We implemented this for all 50 Meta-World environments as well as Throwing and Picking.

**Codebases: NDPs (ours) and Baselines**

Our code can be found at: `https://shikharbahl.github.io/neural-dynamic-policies/`. Our algorithm is based on top of Proximal Policy Optimization (PPO) [4] from `https://github.com/ikostrikov/pytorch-a2c-ppo-acktr-gail` [2]. Additionally, we use code from Whitney et al. [5] (DYN-E): `https://github.com/willwhitney/dynamics-aware-embeddings`. For our implementation of VICES [3], we use the controllers provided them in `https://github.com/pairlab/robosuite/tree/vices_iros19/robosuite` and overlay those on our environments.

### 1.3 Differentiability Proof of Dynamical Structure in NDPs

In Section 3.2, we provide an intuition for how NDP is incorporates a second order dynamical system (based on the DMP system, described in Section 2) in a differentiable manner. Let us start by observing that, when implementing our algorithm, $y_0, \dot{y}_0$ are known and $\ddot{y}_0 = 0$, as well as $x_0 = 1$. Assuming that the output states of NDP are $y_0, y_1, ..., y_t, ...$ and assuming that there exists a loss function $L$ which takes in $y_t$, we want partial derivatives with respect to DMP weights $w_i$ and goal $g$:

$$\frac{\partial L(y_t)}{\partial w_i}, \quad \frac{\partial L(y_t)}{\partial w_i} \tag{1}$$

$$\frac{\partial L(y_t)}{\partial y_t} \frac{\partial y_t}{\partial w_i} \tag{2}$$

Starting with $w_i$, using the Chain Rule we get that

$$\frac{\partial L(y_t)}{\partial w_i} = \frac{\partial L(y_t)}{\partial y_t} \frac{\partial y_t}{\partial w_i} \tag{3}$$

Hence, we want to be able to calculated $\frac{\partial y_t}{\partial w_i}$. For simplicity let:

$$W_t = \frac{\partial y_t}{\partial w_i} \tag{4}$$

$$\dot{W}_t = \frac{\partial \dot{y}_t}{\partial w_i} \tag{5}$$

$$\ddot{W}_t = \frac{\partial \ddot{y}_t}{\partial w_i} \tag{6}$$

From section 3.2 we know that:

$$\ddot{y}_t = \alpha(\beta(g - y_{t-1}) - \dot{y}_{t-1} + f(x_t, g) \tag{7}$$

and, the discretization over a small time interval $dt$ gives us:

$$\dot{y}_t = \dot{y}_{t-1} + \ddot{y}_{t-1}dt, \quad y_t = y_{t-1} + \dot{y}_{t-1}dt \tag{8}$$

From these and the fact that $y_0$, $\dot{y}_0$ are known and $\ddot{y}_0 = 0$, as well as $x_0 = 1$, we get that $y_1 = y_0 + \dot{y}_0 dt$ and $\dot{y}_1 = \dot{y}_0 + 0dt = \dot{y}_0$, as well as $\ddot{y}_1 = \alpha(\beta(g - y_0) - \dot{y}_0 + f(x_1, g)$.

Using Equations (7) and (8) we get that:

$$W_t = \frac{\partial}{\partial w_i}(y_{t-1} + \dot{y}_{t-1}dt) \tag{9}$$

$$W_t = W_{t-1} + \dot{W}_{t-1}dt \tag{10}$$

and

$$\dot{W}_{t-1} = \dot{W}_{t-2} + \ddot{W}_{t-1}dt \tag{11}$$

In turn,

$$\ddot{W}_{t-1} = \frac{\partial}{\partial w_i}(\alpha(\beta(g - y_{t-2}) - \dot{y}_{t-2}) + f(x_{t-1}, g)) \tag{12}$$

From section 3.2, we know that

$$\frac{\partial f(x_{t-1}, g)}{\partial w_i} = \frac{\psi_i}{\sum_j \psi_j}(g - y_0)x_{t-1} \tag{13}$$

Hence:

$$\ddot{W}_{t-1} = \alpha(\beta(-W_{t-2}) - \dot{W}_{t-2}) + \frac{\psi_i}{\sum_j \psi_j}(g - y_0)x_{t-1} \tag{14}$$

Plugging equations the value of $\ddot{W}_{t-1}$ into Equation (11):

$$\dot{W}_{t-1} = \dot{W}_{t-2} + (\alpha(\beta(-W_{t-2}) - \dot{W}_{t-2}) + \frac{\psi_i}{\sum_j \psi_j}(g - y_0)x_{t-1})dt \tag{15}$$

Now plugging the value of $\dot{W}_{t-1}$ in Equation (10):

$$W_t = W_{t-1} + (\dot{W}_{t-2} + (\alpha(\beta(-W_{t-2}) - \dot{W}_{t-2}) + \frac{\psi_i}{\sum_j \psi_j}(g - y_0)x_{t-1})dt)dt \tag{16}$$

We see that the value of $W_t$ is dependent on $W_{t-1}, \dot{W}_{t-2}, W_{t-2}$. We can now show that we can acquire a numerical value for $W_t$ by recursively following the gradients, given that $W_{t-1}, \dot{W}_{t-2}, W_{t-2}$ are known. Since we showed that $y_0, \dot{y}_0, y_1, \dot{y}_1$ do not require $w_i$ in their computation, $W_1, \dot{W}_0, W_0 = 0$. Hence by recursively following the relationship defined in Equation (16), we achieve a solution for $W_t$.

Similarly, let:

$$G_t = \frac{\partial}{\partial g}(y_{t-1} + \dot{y}_{t-1}dt) \tag{17}$$

$$G_t = G_{t-1} + \dot{G}_{t-1}dt \tag{18}$$

and

$$\dot{G}_{t-1} = \dot{G}_{t-2} + \ddot{G}_{t-1}dt \tag{19}$$

Using section 3.2, we get that

$$\frac{\partial f(x_{t-1}, g)}{\partial g} = \frac{\psi_j w_j}{\sum_j \psi_j} x_{t-1} \tag{20}$$

Hence:

$$\ddot{G}_{t-1} = \alpha(\beta(1 - G_{t-2}) - \dot{G}_{t-2}) + \frac{\psi_j w_j}{\sum_j \psi_j} x_{t-1} \tag{21}$$

and we get a similar relationship as Equation (16):

$$G_t = G_{t-1} + (\dot{G}_{t-2} + (\alpha(\beta(1 - G_{t-2}) - \dot{G}_{t-2}) + \frac{\psi_j w_j}{\sum_j \psi_j} x_{t-1})dt)dt \tag{22}$$

Hence, $G_t$, similarly is dependent on $G_{t-1}, \dot{G}_{t-2}, G_{t-2}$. We can use a similar argument as with $w_i$ to show that $G_t$ is also numerically achievable. We have now shown that $y_t$, the output of the dynamical system defined by a DMP, is differentiable with respect to $w_i$ and $g$.

## 1.4 Ablations

We present ablations similar to Figure **??**, using the Throwing task. The results are showin in Figure 1. We see that NDPs show similar robustness across all variations. Secondly, we ran ablations with the forcing term set to 0 and found it variant to be significantly less sample efficient than NDPs while converging to a slightly lower asymptotic performance. Finally, we ran ablation where $\alpha$ is also learned by the policy while setting $\beta = \frac{\alpha}{4}$ for critical damping. We see in Figure 2 that NDPs outperforms both settings where $\alpha$ is learnt and where the the forcing term $f$ is set to 0.

Additionally, we ran multiple ablations for the VICES baseline. We present a version of VICES for throwing and picking tasks that acts in end-effector space instead of joint-space (we call this 'vices-pos')., as well e ran another version of VICES where the higher level policy runs at similar frequency as NDP which we call 'vices-low-freq'. The results are presented in Figure 2a and Figure 2b. We found it to be less sample efficient and have a lower performance than NDP.