[Reviews · NeurIPS 2020]

Review 1

Summary and Contributions: The authors introduce Neural Dynamics Policies (NDPs), a novel trajectory action space for reinforcement and imitation learning where the policy outputs the parameters of a dynamical system and this dynamical system outputs actions via forward integration. This process is fully differentiable and the authors demonstrate its use in both reinforcement learning and imitation learning, outperforming several baselines.

Strengths: The authors propose a novel action space that is different from other works in the field of action spaces for reinforcement learning by leveraging Dynamic Movement Primitives (DMPs), which have traditionally been used for imitation learning. This formulation is clever and has several benefits - including temporal abstraction, reasoning about trajectories in a low dimensional action space, and the authors demonstrate strong empirical results showing improved sample efficiency and performance against a number of baselines. Furthermore, the policy is able to make decisions at a lower frequency than the control frequency of the system, since once the dynamical system parameters are chosen, the system can simply be evolved forward in time open-loop for a number of time steps. This is useful for real-world settings where the policy needs to run in real-time without costly inference.

Weaknesses: The biggest limitation is the lack of imitation learning experiments. The authors chose to conduct an imitation learning experiment in a digit writing domain. However, the authors ran extensive RL experiments in a variety of robotic manipulation domains - I strongly advise using a policy trained on these domains as an expert (preferably not trained with NDP) and running behavioral cloning experiments using NDP and comparing against other action spaces and policy architectures. This would help decouple the benefit of NDP for exploration from the benefit of NDP as an action representation for control and modeling action sequences and should be a fairly straightforward experiment to run. It would also be interesting to see the effect of the control frequency and subsampling expert action sequences in the data - something that NDP is uniquely suited to do. For robotic manipulation domains, it would be interesting to provide more comparisons to other choices of action space (for example, operational space / end-effector control, as done in VICES, instead of pure joint space control). The ablation experiments could also be improved. Figure 6 provides ablations on the Pushing task, but Figure 5 shows that many baselines are performant here - it would be interesting to provide ablation results for something like the Throwing task, since NDP outperforms other methods by a wide margin here. Also, the VICES baseline is flat - this is questionable - while the authors claim the reason of large action space dimensionality, a fairer comparison could be provided by running VICES at a lower policy frequency as NDP does (e.g. output the goal joint position and the impedance parameters every k steps). Another important ablation, which could be easily run in an imitation learning setup, is to evaluate the effect of learning different parts of the dynamical system. For example, is it useful to also learn alpha and beta in eqn 4? Does it suffice to just learn g and set the forcing term to 0? Update (post-rebuttal): The authors ran additional imitation learning experiments, a fairer comparison for VICES, and provided additional ablations. They promised to include all results in the camera-ready, satisfying all of my concerns.

Correctness: The empirical results are impressive and support the claims made by the authors - the only major concern is the lack of Imitation Learning experiments in the same domain as the RL experiments. Update (post-rebuttal): The authors have now included more imitation learning experiments.

Clarity: The paper is mostly well-written, but the clarity of the paper could be improved in some sections. Readers would benefit from a more careful explanation of DMPs in Section 2, including intuition behind equations 1-3, especially the forcing term. There are a few typos in equations that impede understanding at times. The description for PPO in section 3.4 is very unclear - this needs to be made more clear. The algorithm block is also confusing - the authors should make the difference between k (NDP rollout length) and M (number of integration steps) more explicit. My best guess right now is that multiple integration steps take place per environment action step but this is not clear from the algorithm block. Additional details on the "g" function used in the robot manipulation experiments would be good to include.

Relation to Prior Work: Prior work is adequately discussed.

Reproducibility: Yes

Additional Feedback: A few typos and comments follow: Line 84 - typo - a_x controls rate of decay Line 86 - using i as an index is extremely confusing - the coefficients seem like complex numbers Line 111 - g is a poor choice of notation for the inverse controller, as it is also used to denote the goal in Section 2 Line 136 - "frequence" Line 146-147 - sentence missing conjunction, like "and" Line 169 - this is confusing 1-5 hz is actually slower than 100 hz, so decisions are being made less frequently Equation 4 - missing a parenthesis Videos in supplemental - indeed NDP is qualitatively smoother than PPO, but it is also rather curious that for the soccer task, the NDP policy seems to waste some actions before starting to correctly hit the ball each episode.


Review 2

Summary and Contributions: The authors are starting from the observation that current approaches for end-to-end learning of robotic behaviors (whether it is imitation or RL based) directly generate raw actions, such as torque, joint angle or end-effector position. In contrast, classical robotics had often reasoned about dynamic trajectories parameterized in such a way as to capture the physical constraints of the movement. An example of such approach is dynamic movement primitives (DMP), which the authors are using as a subcomponent in their model (although other approaches are possible). The authors are proposing to combine the flexibility of end-to-end learning with the dynamic models of classical robotics. In particular, the proposed approach generates the parametrization w and the goal g of a force function - effectively embedding the dynamic structure in the last layer of deep neural network. As the dynamic system represented in the last layer is itself differentiable, the overall system allows for end-to-end learning the standard techniques used in deep learning. In the case of imitation learning, the backpropagated loss is the standard behavior cloning loss in the form a squares of target and policy differences. For the reinforcement learning model, the authors are using PPO as the deep RL algorithm. Due to the way in which the last step of the system is controlling a dynamic model, the network outputs one value for every k environment step. The resulting approach is evaluated in simulation over a variety of tasks.

Strengths: The paper is theoretically well grounded, and it is especially strong in its integration of deep learning and control theory. The empirical evaluation is standard for the type of work (OpenAI Gym / MuJoCo). The proposed approach is particularly important because, as the authors correctly point out, end-to-end learned neural network behaviors perform worse than classical robotics approaches in situations where the dynamics of the robot and environment is a major factor.

Weaknesses: It would be better if the authors clarify the way they are using the proposed approach for reinforcement learning (section 3.4). It seems that the reward is coming from a comparison with a desired trajectory - but this is not explicitly stated. The authors are showing the loss explicitly for imitation learning (eq 7) but no similar term is presented for the policy gradient. We agree with the author's criticism of current approaches not taking into account the dynamics of the robot / environment when the output is torque or joint angle. We want to point out, however, that if the output is an end-effector position (eg a waypoint), dynamics can be integrated in a natural way. That is, the approach proposed by the authors is not the only posibility, and I would have liked more discussion of the differences from such an approach.

Correctness: As far as I can tell, the claims, method and evaluation approach are correct.

Clarity: Overall, the paper is very well written.

Relation to Prior Work: I am not aware of any prior work which overlaps with this paper.

Reproducibility: Yes

Additional Feedback: I read the rebuttal document, which satisfactorily answers a question I posed.


Review 3

Summary and Contributions: This manuscript presents a method to learn to generate robot trajectories (via imitation or reinforcement) using dynamic movement primitives (DMPs) as parametrization. DMPs are a well known trajectory representation in robotics and attain several benefits (convergence, stability, smoothness…). The work here combines this parametrization with deep neural network architectures by making use of the known derivatives of the DMPs’ generated motions wrt. the parameters to train end-to-end. One of the main novelties is the MPC-like procedure to be able to roll-out the DMP-parameters in a RL setup and back-propagate the error.

Strengths: -Well written: clear, well structured -Theoretically sound -Relevant to the NeurIPS community: novel action representation that can facilitate learning motion for robotics -Extensive empirical evaluation: including imitation learning and reinforcement learning, in various tasks, with multiple baselines. This is really loable -Good results -Includes videos and code Based on the response letter, I have updated my score

Weaknesses: - Novelty: it is unclear how this work is different from “Deep Encoder-Decoder Networks for Mapping Raw Images to Dynamic Movement Primitives” by Pahic et al., at least in the part of imitation learning. The comparison to this work in the related work is clearly insufficient for a work that is, in essence, very much the same concept: learn DMP parameters as action space. Even the derivatives, that look to be presented here as “novel”, are part of that work already, with more details and special cases for SE(3). Please, make much clearer the novel contributions of this work. This includes to clearly state in your explanations (including the ones about DMPs) what is novel and what is taken from prior work and just summarized in the paper. For someone without experience with DMPs it could sound like much of the explained work is part of the contribution while it is just the general DMP formalism. I think there are novel parts in this work, namely the neural architecture and the MPC-like RL procedure, but it is necessary to give correct credit to the authors of the theories this work is based on. - Related work: This part needs some work. The section is superficial and vague. DMPs have been used in robot learning for almost 20 years. It is also not true that DMPs have been only used in imitation learning; they have been extensively used in reinforcement learning setups, especially by Stefan Schaal’s group (Kober, Pastor, Kalakrishnan, Stulp). Please, connect also to the large corpus of work by Aude Billard: she also has been learning dynamical systems for decades, although with different parameterizations. Also, since some of the works are used as baselines, I would explain them deeper (with pros and cons) and bring that discussion sooner, maybe after intro and background, to help the reader understand the results of the experiments.

Correctness: The method is correct. The DMP equations are right. The derivatives seem correct. How do they compare to the derivatives in Pahic et al.?

Clarity: Yes

Relation to Prior Work: No, as explained before this is one of the main weaknesses, which makes it hard to assess the contribution and novelty of this work.

Reproducibility: Yes

Additional Feedback:


Review 4

Summary and Contributions: The paper presents an approach for training robotic policies that predict parameters of a dynamical system instead of raw robot actions. In particular, dynamic movement primitives (DMP) framework is embedded within a deep learning architecture. A high-dimensional neural network predicts parameters of the DMP forcing function, such as basis function weights and the goal, given the current state. Acceleration outputs of the DMP are then integrated by a forward integrator and used to produce actions with an inverse controller. The paper provides derivatives for the single components, such that they can be trained with back-propagation. The framework is employed in imitation and reinforcement learning settings on visual and robotic tasks. It is shown to outperform PPO and other methods that learn to output parameters of robotic controllers such as VICES and Dyn-E.

Strengths: - Representing robot behavior as a dynamical system is a natural way to impose continuous structure onto robot actions. - Presented method allows to vary the inference frequency as the predicted dynamical system can be run for several steps. This is important for combining high-frequency control with low-frequency inference when using high-dimensional inputs, such as camera images. - Experiments demonstrate that the presented method outperforms baselines on visual and robotic tasks. - Presented ablation studies demonstrate the effect of the various parameters of the system.

Weaknesses: - Although the method is tested on a visual digit drawing task, It would be interesting to see more robotic experiments with visual inputs. - It would be interesting to see more real robot experiments to better understand how the method can deal with a lower number of robotic trials and real-world uncertainty.

Correctness: Claims and mathematical derivations in the paper are coherent and empirical evaluation correctly shows performance improvements over the baselines.

Clarity: The paper is well-written and easy to understand and follow.

Relation to Prior Work: The paper clearly establishes connection to prior works and uses them as baselines for the experiments.

Reproducibility: Yes

Additional Feedback: Post-rebuttal comments: Thanks for addressing reviewer's comments and providing new details and experiments in the rebuttal.

[Author Response · NeurIPS 2020]

We thank the reviewers and are glad that they find our work theoretically sound [R2, R3], well-written [R1, R2, R3, R4]
and empirically impressive [R1, R3]. R2 says "the paper is theoretically well grounded, and it is especially strong in
its integration of deep learning and control theory" and R4 finds it "important for combining high-frequency control
with low-frequency inference when using high-dimensional inputs". R1, R4 suggested additional imitation learning
experiments and ablations. *We are pleased to report that we have completed all those experiments.* Due to limited space,
we can't provide all answers and result plots but promise to include them in the camera-ready which allows 9 pages.

[**R1**, **R4**] *"lack of imitation learning experiments... in robotic manipu-*
*lation domains"*: Exactly as R1 advised, we train PPO-based experts on
robotic manipulation tasks from our paper and use it to collect data which
is then used for learning NDPs and a baseline neural network policy via
behavior cloning. Success rates in Table 1 show superior performance for
NDPs.

| Method | NN | NDP (ours) |
|--------|-----|------------|
| Throw  | $0.528 \pm 0.262$ | **$0.642 \pm 0.246$** |
| Push   | $0.002 \pm 0.004$ | **$0.208 \pm 0.049$** |
| Soccer | $0.885 \pm 0.016$ | **$0.890 \pm 0.010$** |
| Faucet | $0.532 \pm 0.231$ | **$0.790 \pm 0.059$** |

Table 1: Imitation learning on robotic tasks. Success rate is out of 1.

[**R1**]*"a fairer comparison could be provided by running VICES at a*
*lower policy frequency"*: As suggested, we ran another version of VICES
where the higher level policy runs at similar frequency as NDP. Results indicate that *NDP still outperforms it by approx.*
*75%.* We can't include the plots here due to space constraint but promise to include them in the camera-ready.

[**R1**] *"ablation results for... the Throwing task.", "is it useful to also learn alpha and beta in eqn 4?", "Does it*
*suffice to just learn g and set the forcing term to* 0*?"*: We have finished all these three set of experiments. Firstly,
upon running similar ablations as pushing for throw task too, we found that NDPs show similar robustness across all
variations. Secondly, we ran ablations with the forcing term set to 0 and found it variant to be significantly less sample
efficient than NDPs while converging to a slightly lower asymptotic performance. Finally, we ran ablation where $\alpha$ is
also learned by the policy while setting $\beta = \frac{\alpha}{4}$ for critical damping. We found it to be less sample efficient and have a
lower performance than NDP. We will include all plots corresponding to all these settings in the final version.

[**R1**, **R2**] *"The description for PPO... needs to be made more clear", "clarify the way... the proposed approach [is*
*used] for reinforcement learning", "no [explicit loss] term is presented for the policy gradient"*: We will update the
final version to include a separate paragraph about our modifications to PPO and it's objective function. We will also
clarify in NDP-RL algorithm block that $M$ integration steps are used in each of the $k$ rollout steps. Thank you.

[**R2**] *"if the output is an end-effector position (eg a waypoint), dynamics can be integrated in a natural way."*: This
is a good point. In fact, the approach described in VICES creates waypoints (e.g. a cubic spline) in end-effector or joint
space. To get natural or smooth outputs, such methods often require tuning (for every task), or employ high dimensional
actions (such as VICES). NDPs are less susceptible to these issues because the final outputs are low level actions.

[**R3**] *"DMPs have been used... for almost 20 years...Stefan Schaal's group. connect also to the large corpus of*
*work by Aude Billard"*: Indeed, we are inspired by the large body of work on DMPs from Stefan Schaal's, Jan Peter's
and Aude Billard's group. We believe our work is complimentary as it bridges the gap between recent advances in
end-to-end deep RL and seminal work on dynamical systems. Unlike our end-to-end architecture, most prior works
use either a single DMP to represent the whole trajectory or the trajectory is manually segmented to learn different
DMPs. In the current version of our manuscript, we already start the introduction by mentioning these seminal papers.
To further pay due respect to the two decades of literature, as suggested by R3, we will expand the discussion of these
papers and reorganize accordingly.

[**R3**] *Differences from Pahic et al.*: Pahic et al. only tackle imitation learning from demonstrations. This is because they
use only one DMP to represent the whole task trajectory. In contrast, our NDP framework can output a new dynamical
system for each timestep to fit diverse trajectory behaviours over time. Hence, NDP can easily be incorporated not just
in imitation learning but also end-to-end deep RL setup. We do not claim that DMP formalism itself is our contribution
(also agreed by R1,R2,R3) and will further make it absolutely clear in Section 3.2 where derivatives are discussed. We
also provided a detailed proof in the appendix by showing a recurrence relationship between partial derivatives.

[**R3**] *"I would explain [baselines] deeper (with pros and cons)..."*: This is a great suggestion! We will update final
version to include a discussion on the action dimensionality & impedance controllers used in VICES, as well as a
discussion of DynE's dimensionality reduction, temporally abstract actions and its tendencies to overfit to passive data.

[**R3**] *"I am inclined to increase my score if the weaknesses are fixed"*: We sincerely hope the concerns are addressed.

[**R4**] *Imitation learning experiments*: Please see first answer on top.

[**R4**] *"It would be interesting to see more real robot experiments"*: We are indeed planning to conduct real robot
experiments for throwing and other tasks. However, due to COVID-19, we did not have access to the lab and were
unable to run real robots prior to the deadline. We hope to have robot results ready before the camera-ready deadline.

[Meta-Review · NeurIPS 2020]

The paper proposes a very interesting, novel policy representation with extensive evaluations both for imitation learning and reinforcement learning. The reviewers highly appreciated the additional insights and experiments in the rebuttal.